# Integration and Investigation of Selected On-Board Devices for Development of the Newly Designed Miniature UAV [note 1]

**DOI:** 10.3390/s20061752

**Published:** 2020-03-21

**Authors:** Aleksander Olejnik, Robert Rogólski, Łukasz Kiszkowiak, Michał Szcześniak

**Affiliations:** Faculty of Mechatronics and Aerospace, Military University of Technology, Warsaw 00-908, Poland; aleksander.olejnik@wat.edu.pl (A.O.); lukasz.kiszkowiak@wat.edu.pl (Ł.K.); michal.szczesniak@wat.edu.pl (M.S.)

**Keywords:** miniature UAV, autopilot, on-board sensors, miniature propulsion

## Abstract

The article is a development of the topic generally presented in the conference proceedings issued after the 2019 IEEE International Workshop on Metrology for Aerospace. In contrast to topics presented in the conference, the article describes in detail avionic equipment and on-board systems integration process and their in-flight adjustment in regard to the newly designed miniature unmanned aerial vehicle (mini-UAV). The mini-airplane was constructed and assembled in the course of the research project, the purpose of which was to show implementation of a totally new mini-UAV design. The intention of the work was to develop a new unmanned system including an originally constructed small airplane with elements purchased from open market. Such approach should allow to construct a new aerial unmanned system, which technologically would not be very advanced but should be easy to use and relatively inexpensive. The demonstrator mini-airplane has equipment typical for such an object, i.e., electric propulsion, autopilot system, camera head and parachute device for recovery. The key efforts in the project were taken to elaborate an original but easy to use system, to integrate subsystem elements and test them so to prove their functionality and reliability.

## 1. Introduction

Miniature unmanned aerial vehicles are very small reconnaissance flying objects of which the total weight according to widely known categorization documents is approximately between 5 and 30 kg, wherein stated definitions for this type of UAV are rather convergent both from the side of responsible civil and military organizations [1,2]. Fragment of the official classification presenting the position of miniature objects is shown in Table 1. Due to their small size (usually adequate for large flying models) and light payload, they are not a very expensive assortment of products offered by the aviation industry. Not very high technological requirements, general availability of construction, drive and equipment components, low price of both components and the final product–these all are reasons which make the use of mini-UAV more and more common and justified in various fields of activity. In military applications, a mini-plane equipped with a TV camera and silent electric drive is an excellent recognition instrument for the platoon/company level subdivision or an independent special group. There are many cameras dedicated to this type of mission. However, in addition to strictly military applications, mini-UAVs are widely used in civilian enterprises related to the activities of both state institutions and private enterprises.

A significantly crucial problem during the development of a new unmanned aerial system (UAS) is the integration of all devices mounted in a miniature airplane. To assure full system complexity and, in consequence, required functionality, this integration must be fulfilled at the following three levels: mechanical assembly, electric connectivity and achieving readiness for operation of each subsystem. The mini-UAV described here and previously presented more generally [3] was developed as a demonstrating product of some R&D projects carried out to prove usability and effectiveness of a small flying platform designed and constructed for the purposes of video monitoring and inspecting in close range and at low altitude. The primary goal was to design, construct and test a few technology demonstrators of mini-UAV obligatory equipped with the following devices or subsystems:electric propulsion, with an electric engine, pulling propeller and controller;automatic flight control system, ensuring a stable flight on a given route in autonomous mode (commercially offered autopilot MP 2128g was applied in the project);camera gimballed head, enabling day and night terrain recognition (in TV or thermal mode);communication subsystem, ensuring transmission of telemetry and image data;recovery system, with automatically ejected parachute;power supply system with electric sources and wiring.

UAS system components are subjects of numerous researches and tests described in wide bibliography, available in electronic or paper forms; exemplary items [4,5,6,7,8,9,10] are worth citing. In the presented project challenges, one can distinguish certain trends of activities determining the success of the finally developed system. These lines of action can be divided as follows:-aerodynamic-and-structural airframe design and construction;-selection of components and optimization of the propulsion;-material and manufacturing technologies;-architecture of avionic on-board subsystems, including devices of flight control and stabilization, navigation, recognition, communication, self-recovery and others;-ground control station architecture;-mechatronic integration and system validation tests;-research and qualification tests;-developing optimal operating and maintenance procedures.

Each action is taken at a specific stage of system development, and each stage defines specific technical or technological tasks. However, only meeting a set of requirements at all development levels together can contribute to achieving the final success, which is the development of a fully usable and commercially available product.

The problems presented in the paper were chosen as the most complicated and demanding from the range of all development activities undertaken towards developing a new system. Specific elements of the abovementioned equipment were being sequentially mounted on mini-airplane and then tested separately—at first during on-ground investigations and then during flight-tests. The target pre-prototype aircraft was equipped with all possible subsystems. It was fully completed and intended for use in the mode of supervised trial operation. Specific aspects of the UAV construction, equipment adaptation and its testing were described here. Particularly important measurements taken to verify operational readiness of deployed devices were carried out in the context of the integration of the autopilot system and the selection of the power unit components.

## 2. UAV Design

Mini-UAV Rybitwa (*eng. Tern*) of which CAD designs are presented in Figure 1 is a miniature made-of-composite airplane with traditional aerodynamic outline (high-wing monoplane with T-tailed boom). The lifting plane is assembled with 4 semi-monocoque segments and is equipped with ailerons and flaps. The central sections are connected to the fuselage by putting them over rigid carbon girder spar. There is a parachute recovery set inside the fuselage bay. The propulsion system consists of three-phase electric motor and two- or three-blade tractor propeller. The on-board avionic equipment (MP2128g autopilot and gimballed video sensor) are assembled inside the under-fuselage pod. The airframe configuration in a classic arrangement with a suspended slender container has been successfully used in such constructions as: Skylark (Elbit Systems), Skyblade (ST Aerospace) or experimental Kiwit (ATE Technology). Fundamental dimensions and performances achieved in case of the final UAV configuration shown in Figure 2 are the following:

The recognition system distinguishes an air component, i.e., an unmanned aerial vehicle and a ground component comprising a ground control station (GCS) with an antenna terminal. Connectivity between telemetry data, image recognition and remote-control signals is provided between the components. The main components of GCS are portable industrial computer (with a joystick), antenna terminal, transceiver modules, portable source of electric supply. Conceptual view of a system division is presented in Figure 3.

## 3. Autopilot Components and Control System Integration

The MP2128g autopilot—components of which are described in [11,12]—is dedicated to stabilizing and controlling UAVs of various types and sizes, i.e., to enable steady flights in fully autonomous mode. It is possible thanks to many on-board sensors included in the set supporting both start or landing and air cruise navigated in accordance up to 1000 waypoints. The device enables operating in different modes by maintaining such navigational parameters like altitude, airspeed, heading in reference to GPS/INS data and turn coordination. Using additionally added magnetometer, it is possible to estimate the wind correction vector especially needed at crosswind flight.

MP2128g was utilized in many UAV projects with success, which was described in papers like [13,14,15]. In those and many other papers, the challenge related to autopilot application is presented as a purely technological problem or as an original scientific task. The case of typically technological problem occurs, when the new UAV construction is ready and is to be equipped with a dedicated mission equipment, and at the same time, the priority commercial requirement is to deploy an available and reliable control and navigation system. In such a situation, the application of MP2128g is perhaps the best possible choice. The UAV equipped with that autopilot can be investigated in a virtual simulating environment [15], or directly in real flight tests [13]. On the other hand, the leading challenge could be the development of a quite new control system, based on alternative components and original self-elaborated software. For last over a dozen years many designs of own autopilot systems have been developed, which were being validated in the hardware-in-the-loop tests or in real flights [7,16,17]. However, even in those cases, using the commercial autopilot is rather recommended, as it enables to conduct comparison tests or measure flight parameters, which could be some reference data for the one identified by the newly applied set. The type of such dual research is presented in [14]. In this work, parameters recorded by the newly developed autopilot based on the PC/104 module were compared with the MP2000 autopilot parameters, while both devices were mounted simultaneously on the same flying platform.

Although MP2128g is the highly advanced solution, it is anyway not a plug-and-play device, which can be easily integrated with a UAV, especially if the vehicle is newly designed. The most common issues, which must be investigated and solved during integration process, include:Design of vibration isolation system;Components arrangement inside tight mini-airplane fuselage;Selection of radio frequency modems.

The main source of vibration is the engine and a propeller. Vibration generated by these two elements contributes to increasing errors of IMU, especially vibration increases gyroscope bias. An incorrectly selected vibration system may cause gyroscope saturation, which can lead to UAV crash. Since the airframe structure is relatively stiff and the imbalance of rotating masses in the electric drive is also small, autopilot vibro-isolation system was put together in the form of a passive system of four silicone cushions, on which the autopilot was mounted. The suspension, despite the small dimensions of the insulating supports used, turned out to be sufficiently rigid in relation to the deposited mass (50 g). During subsequent flight tests, no malfunctions were noted due to structure vibrations, and recorded flight accelerations did not exceed the value of 1 g. What is more important, gyroscope output was always stable. Figure 4 presents final isolator setup, on the left, and one of the initially tested ones, on the right. As can be seen on the left graph, gyroscope output was reduced almost twice comparing to the data on the right graph—these results were achieved only by changing isolators stiffness. Achieved results were also confirmed by calculating some statistic data, which shows that standard deviation for gyroscope data presented on the left is only 1.46 [deg/s] while standard deviation for gyroscope data presented on the right graph is up to 6.55 [deg/s].

Component arrangement inside tight mini-plane fuselage is not an easy task. Many components like GPS receiver, compass module, radio modem or autopilot include embedded microprocessors or memories which work at high frequency. In these conditions even wrongly led cables can cause interferences between these components. For example, GPS antenna cable placed on the top of GPS receiver module caused sudden increase of GPS jamming. Compass cable, through which SPI signal is sent, placed closed to GPS receiver also caused increase of GPS jamming. Radio modem placed too close to autopilot board caused all uncontrolled servos jerks. The same issue was observed when miniature diversity module was placed too close to PCB servo board. One of Rybitwa requirements was to send real-time video from onboard camera. This requirement caused additional radio modem installation onboard the UAV. Having three radio frequency modules onboard small UAV required installation of three antennas in a manner that did not cause mutual interference. Many tests with spectrum analyzer were done to find the best antenna location.

Items of the autopilot set applied in the project are specified in the Figure 5. The primary ones of them are the following: main core (40 × 100 mm, 28 g), servo expansion board (40 × 39 mm, 18 g), ultrasonic AGL sensor (95 × 55 mm, 92 g), magnetometric sensor (43 × 13 mm, 29.5 g), GPS antenna (38 × 34 mm, 40 g), 2.4 GHz radio modem for telemetric transmission and miniature RC digital servomecanisms (Hitec HS-5125MG).

Before adjusting autopilot to control UAV autonomously, it is necessary to check mini-airplane flight qualities in manually controlled operating, i.e., in RC mode. The flying platform should be capable to hold stable and controllable flight in any possible weather conditions. Only after verifying dynamic capabilities and performances and fixing accepted weights and balance shifts, the integration process can be initialized. In the state of free and stable flight, servos should be configured in such way to assure trim conditions with all control surfaces set in neutral positions (i.e., at zero deflection angles). Demonstration of a tendency towards static stability in the phase of manually controlled flights opens the prospect of testing success in the later autopilot testing phase. Control diagrams of RC mode and mixed (PIC-CIC) mode applied in subsequent flight-test stages are presented in Figure 6 and Figure 7.

After installing components and adjusting controls to hold conditions of aerodynamic equilibrium, there is a necessity to execute flights with autopilot working in flight recording mode. Having verified registered data and assuring that control system modified in accordance to a new architecture (Figure 7) acts properly, the new testing stage could be initialized. During the flight in semi-autonomous mode, the autopilot controls movement and position parameters in accordance with implemented control principles. Any possible state of flight (cruising, climbing, descending, turning or other maneuvers) is maintained by movements of control surfaces (indicated in Figure 8) as effect of combination of some feedback loops. In case of MP2128g, there are the nine basic control loops based on tuned PID controllers [11]:
Aileron from roll–controls ailerons deflection so that the difference between the currently estimated roll value and the required value is as small as possible;Elevator from pitch—controls elevator deflection so that the difference between the current pitch value and the required pitch value is as small as possible;Rudder from –*y* accelerometer—controls rudder deflection so that the difference between the current relative-to-*y*-axis acceleration value and the required–y acceleration is minimalized;Rudder from heading—controls rudder deflection so that the difference between the current heading angle value and the required heading value is as small as possible;Throttle from airspeed—controls throttle set so that the difference between the current airspeed value and the required airspeed value is as small as possible;Throttle from altitude—controls throttle set so that the difference between the current altitude value and the required altitude is as small as possible;Pitch from airspeed—controls elevator deflection (pitch as effect) so that the difference between the current airspeed value and the required airspeed is as small as possible;Roll from heading—controls roll (ailerons and rudder deflection as effect) so that the difference between the current heading angle value and the required heading is as small as possible;Pitch from descent—controls pitch (elevator deflection as effect) so that the difference between the current descent rate and the required descent rate is as small as possible.

The system with active autopilot enables autonomous object control with the possibility of current correction of PID controller settings. However, it is still possible at any time to take manual control over the mini-plane by switching into radio receivers ensuring transmission of the RC signal.

System items were attached inside the mini-airplane in way shown in Figure 9. In case of applying magnetometer, the sensor should be separated from any sources of electromagnetic emissions such as high voltage wires or motor coils. The GPS antenna cannot be shielded. Both the autopilot core and radio modem require DC power supply from the range of 5 up to 27 V. It is rather necessary to apply separate power sources for autopilot and servomechanisms. Servos could consume large quantities of electricity in short time periods, which could interrupt autopilot operation by sudden voltage drop. General method and selected specified techniques useful by configuring system elements and adjusting controller gains were taken from works like [5,13,14,18,19,20].

All electronic items should be securely fastened and properly connected with wires or with silicone pneumatic hoses. This applies strictly to pressure ports. Distribution of electronic components inside the under-fuselage pod is presented in Figure 10. Inside the storage, there is a carbon plate with installed telemetric radio-modem, autopilot core embedded on vibro-isolators (inside the glass-fiber box) and servoboard module. Li-Po battery packages are suspended under the plate. Inside the hemispherical rear chamber (on the left), there are radio-modem antenna and magnetometer installed. Inside the analogous front chamber (on the right), there are datalog DB-9 connector and the 2.4 GHz RC receiver. Both tip hemispherical chambers are attached to the cylindrical part with Velcro straps. Disassembling the plate with equipment is quite easy; therefore, recharging or replacing batteries is not a complicated service activity for users.

## 4. System Verification in Flight Tests

After installing autopilot set series of verification, flight tests begun. Strict safety procedures must be followed during both preparations and flight operations. Securing and supervising a separate flight zone is absolutely necessary. It is necessary to take into account the terrain, weather conditions and the possibility of collision with other flying objects. Especially hazardous are those ones flying illegally using prohibited frequencies or generating very high-power level, which can cause interferences with UAV equipment. Therefore, the doubled RC-transmission with two separate frequencies served by on-board miniature diversity module was deployed (marked in Figure 5, Figure 6) just to minimize the possibility of jamming the currently emitted control signal.

The task of tuning the autopilot is carried out during a series of flights with switching from the manual control mode (PIC) to the computer in command mode (CIC) during short flight sections. This is because autopilot controls UAV using 13 PID feedback loops, and each of them has to be adjusted separately in an iteration process. In the following flights, the autopilot commands are changed to allow all of the feedback loops to be tuned one after the other. After successful regulation of the loop, the drone should be able to perform on autonomous flights.

### 4.1. Adjusting Accessible Feedback Loops

As was mentioned in one of the previous sections, despite MP2128g autopilot being already integrated with many fix-wings, it is still not a plug and play device. MP2128g controls the UAV using a series of PID feedback loops. The term feedback loop refers to any mechanism that controls a system by adjusting the input of the system based on the measured output of the system; that is why for the newly designed UAV, which is Rybitwa, all 13 PID loops have to be adjusted in real flights. For instance, aileron from roll feedback loop controls the ailerons to minimize the difference between desired roll and actual roll; elevator from pitch feedback loop controls the elevator to minimize the difference between actual pitch and desired pitch, heading from crosstrack error feedback loop controls the desired heading to minimize the distance between UAV and the line defined by the previous waypoint and the next waypoint. These are only three of them, and each gain has the ability to define the gain schedule based on the current airspeed. What is more, output of some of the feedback loops feeds the input of other feedback loops, and autopilot can enable and disable feedback loops during the course of a flight in order to accomplish its user-defined mission. Figure 11 shows which feedback loops are enabled by autopilot during level flight.

Among feedback loops, one can distinguish the two following sorts: inner loops and navigation loops. The inner loops are the primary stability loops and must be tuned before proceeding to navigation loops. Navigation loops are dependent upon ground speed, and they require gain scheduling because UAV is equipped with a slower update rate GPS. Typical test flight conducted under mixed control, that means switched between the PIC and CIC modes, is being started always at PIC mode. As was mentioned before, in the first stage, it is necessary to set gains firstly for priority loops, i.e., elevator from pitch, aileron from roll, and rudder from lateral acceleration. These are internal loops, the most important in terms of providing basic flight stability what is explained in References [5,11,13,20,21].

The pClimb command declared in file *.fly file displayed in Figure 12 below is dedicated to maintaining constant pitch angle while entering the targeted ceiling attempting to hold the roll and the sideslip at 0 values. With “climbpitch” parameter set to zero, the drone stops climbing, so the “pClimb” command would not allow to exceed the limit altitude of 9994 m. Parameter “thoverride” set to 0 enables the user to control the throttle manually. This procedure allows to adjust gains in flights with or into the wind for any speed ranges.

Feedback loops are regulated by the PIC controllers’ settings. Proper gain values can be modified in the *.vrs file after each flight. However, there is also some other method to adjust gains with the use the Horizon Software option called “Status Monitor”. This tool enables the operator to observe flight parameters displayed on graphs in the real time and tune PID gains directly if necessary. The modified data can be transmitted to the autopilot via telemetric link, and the UAV stays in further control in accordance with actually tuned loops. However, that data is available only temporarily; therefore, in-flight gain changes must be copied from autopilot RAM to the *.vrs file just after landing.

In Figure 13, Figure 14 and Figure 15 showing runs of flight trajectory and of specific parameters curves of PIC mode parameters are marked in red and sections referring to CIC mode are marked in blue. The CIC flight heading was set close to the wind direction to avoid possible wind affect onto the basic flight stability. When the drone was flying in the CIC mode, all angle values had to be visually tracked to assure the stabilization is properly maintained, i.e., without oscillations. At the same time, in real-time, P, I, D terms and current and desired roll and pitch signals are displayed and monitored on GCS console by the operator. If the plane’s nose seems to oscillate up and down and then gain values for the loop, elevator deflection from pitch had to be decreased in real-time. Similar actions were taken in respect to other loops: ailerons from roll and yaw from lateral acceleration. To achieve better UAV turns without the aircraft nose dropping and adverse yaw, additional term called “feed forward” was set for three feedback loops responsible for control of the rudder based on Y accelerometer data, control of the elevator to minimize the difference between actual pitch and desired pitch and control of the throttle to maintain desired aircraft speed.

It is generally noticeable that in the following CIC intervals, the regulation effect was being improved. High values of red curves in Figure 14 and Figure 15 concern flight sections with turnings performed in PIC mode. During flight sections performed in CIC modes runs of current parameters are strictly close to the desired ones.

To estimate quality of loop adjustment, in CIC mode, error between current and desired values were calculated. For instance, for the pitch feedback loop (Figure 16), standard deviation for the time between 120 and 200 s is 0.84 degrees. This result should be considered as good especially if we take into account that there is a delay between current and desired pitch signals.

### 4.2. Testing System in Autonomous Flight

Once the inner loops were adjusted, navigation loops can be investigated to find the best P, I, D terms which allow fully autonomous flights. MP2128g autopilot ensures autonomy of the mission at any kind of flight stage. There are many types of possible takeoff and landing procedures, including winch, launchers, classic runway takeoffs, parachute recovery, deep stall or classic landings. In this study the UAV was started by the hand-throw and continued flight along the preprogrammed track autonomously [11,12].

Using the *.fly file included below, the UAV can make autonomous flight through 4 defined waypoints. The waypoint input format is longer than takeoff point in rectangular coordinate system. There are also three models added with preprogrammed conditions. They are set by the operator in the GCS. The safety procedure should be introduced to enable getting the UAV back to the takeoff area in case of losing RC transmission.

In Figure 17 below, flight path based on 4 arbitrary waypoints is shown. Trajectories crossing corner points of previously defined square field were circled during navigation loop adjustment. Each waypoint can be moved or deleted at any time during the flight, however, to ensure constant conditions for the procedure the constant trajectory was used to allow comparison between each adjustment stage.

On the graphs below (Figure 18 and Figure 19) selected parameter curves are shown—desired and measured values of altitude, heading and roll in the specified period of time during the left turn. Even in case of pretty strong turbulences, controlled values of roll and heading were very close to the desired ones. Estimated heading error occurred was never bigger than 10%. As can be seen in Figure 18, the heading loop was adjusted so that the current heading is slowly keep up the desired values. This desired behavior is because of the onboard electro-optical head. If the head is in tracking mode, sudden changes in heading can cause huge LOS error between target and pointing vectors. Such small UAV has payload capacity limitation, which affects the onboard payload. Limitation of take-off weight or power consumption affect the flight performance as it is in that case.

## 5. Investigations of the Miniature Electric Propulsion Dedicated to UAV

In the aspect of selecting the components of the mini-airplane propulsion system, a series of measurement experiments were carried out. Some dependencies determined after measurements of propulsion qualities allowed to compare various configurations, which differed one from another because of component type. The research method is based on directions described in [9] and [22]. The aim of the study was to determine the following dependencies:
Thrust vs. rotational speed T = f(n);Torque vs. rotational speed M = f(n);Effective power vs. rotational speed P_eff_ = f(n).

Appropriate measurement requires axial mounting of the engine on the measuring testbed. To perform the tests, some special equipment items are required, such as rotational speed controller, PWM pulse generator (JETIBOX), battery packs. In order to determine the abovementioned characteristics, the following measurements are to be performed:
Thrust F [N];Force from the reaction moment R [N];Rotational speed of the propeller n [rpm];Battery voltage level U [V];Current consumed by engine I [A].

The test object was a three-phase GT5325-11 electric motor. The measuring system assembled on the testbed and illustrated in Figure 20 consists of the following items generally available on the market:
-Engine GT 5325-11 with APC 20 × 10 propeller (connected to the controller EMAX ESC-100A-HV);-Miniature datalogger and clamp amperemeter (for measuring and storing current/voltage data);-Force measurement track (for measuring F and R values);-PWM pulse generator;-Voltage supply source for engine (4/8 cells).

The tested engine was mounted on the testbed with the use of specially prepared steel mount. The tested motor is powered by three-phase AC voltage, which requires the use of a special speed controller for brushless motors. The engine speed is controlled by the EMAX ESC-100A HV regulator. The applied controller is adapted to supply from 4 to 10 Li-Pol cells. It is also characterized by a temporary maximum current up to 100 A; this value exceeds by 20% the maximum amount of current that can flow through the stator windings of the tested engine. In order to unambiguously determine the measurement points, a JETIBOX module was used to generate the PWM pulse, which could control the speed controller in the range from 1.024 ms to 2.047 ms. During the tests, the three different power configurations were applied:(1)4 s Li-Pol battery pack − 1 × 14.8 V (5300 mAh/25 C),(2)6 s Li-Pol battery pack − 1 × 22.2 V (5550 mAh/28 C),(3)8 s Li-Pol battery pack − 2 × 14.8 V (5300 mAh/25 C).

The conducted tests enable unambiguous determination of key characteristics for investigated UAV propulsion, i.e., torque, thrust, effective power and efficiency—each parameter versus rpm (revolution per minute rate). Their graphs are shown in Figure 21 and Figure 22.

The aim of the study was to select the most suitable power source for a specific engine and propeller, which means the pre-selected propulsion performance should be enough to assure required thrust (thrust-to-weight ratio is more adequate) and time of flight mission. In the course of the research, a series of measurements were carried out for each of the three available power packages of nominal voltage: 14.8 V (4 s pack.), 22.2 V (6 s), 29.6 V (8 s).

On the basis of the parameters obtained, a group of characteristics was prepared for various battery package configurations. It can be concluded that the mechanical efficiency was the highest for six power cells. Efficiency value is at the level about 70%, and it is sustained in the quite high range, i.e., between 2000 and 5000 rpm. The graph curves generated as an effect of approximation with polynomials of at least third degree do not deviate from the known theoretical runs for similar applications of miniature propulsions. The maximum recorded parameters occurred at the highest measured rpm rate gained at the voltage value of 33.6 V (achieved for 8 s package). Current value does not exceed the threshold 54 A, which allows to conclude the applied speed controller is absolutely sufficient taking into consideration that its maximum acceptable current value is 100 A. The nominal value of the maximum current flowing through the motor winding is about 30 A higher than the maximum recorded value. This means that it is possible to use a larger propeller, which could improve the performance of the tested power unit.

Analyzing thrust and current consumption values for applied voltage sources, it can be concluded that a package consisting of six cells (6 s) fits best to the developed mini-UAV at its maximum take-off weight of about 9 kg—the T/W (thrust to weight ratio) value is in that case about 0.54. Equally acceptable is the 4 s-package, which gave the maximum thrust equal to 24.4 N and in effect the T/W ratio equal to 0.27—value still acceptable for miniature and small UAV. On the other hand, the propulsion system powered with an 8 s-battery seems to be significantly oversized. This configuration is distinguished by the highest level and the largest useful range of the efficiency, but the T/W ratio obtained here is as much as 0.78—much more proper for a combat jet than for a quite slow-flying UAV. Such a high value of the ratio suggests that the propulsion produces too much thrust, which seems to be unprofitable when applied to the object under consideration.

It is necessary to note that the above described tests were carried out in static conditions, i.e., at zero cruising speed. It is known that when the speed of incoming airflow goes up (increased by the term of cruising speed), the efficiency of propellers decreases. Therefore, the above considerations do not reflect the actual description of the phenomena in the full operating range of UAV.

## 6. Concept of Simple Observation Video-Head for Training or Test Flights

For carrying out practice and video transmission tests during flight, a training observation video-head was constructed. The target recognition device is obviously the MicroPilot product: MP Dayview/Nightview. However, to avoid the risk of damaging expensive equipment for the purpose of ad hoc flight tests, a much cheaper and simpler to use device has been developed. The head was equipped with the following components (shown in Figure 23):OEM block camera module—Sony FCB-IX11 AP;Executive servo device HS-5125MG (to drive camera in tilt rotation);On-screen display module—ETS OSD PRO (enabling on-screen projection of any subtitles);Video image transmission module—Immerson 5.8 GHz with 600 mW pure output power;RC receiver—Jeti Duplex R14 (for servo control).

Using the OSD module, it is possible to impose the flight parameters of the mini-airplane on the image transmitted to the ground. The module is directly connected to the TV signal output from the camera. The TV signal is processed by the head video-transmitter and is emitted to the GCS receiver.

The area observed by the head can be changed by tilting the optical axis of the optical sensor in the range of ~30°. This range of camera movement allows novice operators to perform preliminary training flights. The main emphasis in designing this device was concentrated on achieving the same mass in relation to the target head, elaborating simple construction and ensuring minimal possibilities to conduct entry training for flight operators. 

Initially, one can estimate the area that the camera observes depending on the flight altitude of aircraft *H*. Using the geometrical relationships shown in the Figure 24, it is possible determine the depth of field of observation denoted as *b*. Two characteristic angular quantities resulting from the imaging scheme of the monitored area in the plane of the flight path should be taken into account, namely, *τ*, camera tilt angle; *φ*, camera view angle.

The two horizontal distances shown in the scheme can easily be linked to the angles through the appropriate trigonometric relationships:(1)$$a+b=\frac{H}{tg\left(\tau -\frac{\varphi }{2}\right)}$$
(2)$$a=\frac{H}{tg\left(\tau +\frac{\varphi }{2}\right)}$$

After linking both formulas, a function determining the depth of the observed area can be specified:(3)$$b=H\left[\frac{tg\left(\tau +\frac{\varphi }{2}\right)-tg\left(\tau -\frac{\varphi }{2}\right)}{tg\left(\tau +\frac{\varphi }{2}\right)tg\left(\tau -\frac{\varphi }{2}\right)}\right]$$

For the purposes of preliminary calculations, a reasonably estimated camera tilt angle *τ* = 40 ° and a mission ceiling H = 200 m can be assumed. The angle of view is closely related to the focal length *f* and the height of the CCD matrix *h*, which demonstrates the simplified scheme of light projection in lateral view. Based on the scheme (Figure 25), one can determine the viewing angle according to the following formula:(4)$$\varphi =2arctg\frac{h}{2f}$$

After substituting Formula (4) into the initial Formula (3), a more detailed expression for the depth of field observation function can be obtained:(5)$$b=H\left[\frac{tg\left(\tau +arctg\frac{h}{2f}\right)-tg\left(\tau -arctg\frac{h}{2f}\right)}{tg\left(\tau +arctg\frac{h}{2f}\right)tg\left(\tau -arctg\frac{h}{2f}\right)}\right]$$

The Sony FCB-IX11 AP camera mounted in the head device is equipped with a CCD matrix having dimensions of 2.7/3.6 mm (height/width). The camera has a lens with a variable focal length in the range from *f_min_* = 4.1 mm to *f_max_* = 73.8 mm. Using the given relationships, it is possible to determine the approximate depth of visual penetration of the sensor. For the reasonably assumed mission altitude *H* = 200 and the given range of variation *f*, this parameter can vary from *b_min_* = 150 m to *b_max_* = 265 m.

## 7. Conclusions

In the paper, some crucial topics selected from the overall design-and-development process of a new concept mini-UAV are presented. In the whole prototyping process, there are at least five basic stages of system development, namely:
-defining overall conception and specification of technical requirements;-elaborating technical design;-manufacturing and integrating elements and components;-investigating and validating devices or subsystems;-test operating and correcting the whole system.

The problems described here appeared and were being solved during the stages of integrating components and investigating subsystems. The variety of undertaken tasks results from the nature of the research project, which was a typical technological-and-development activity aimed at developing, manufacturing and checking the properties of the unmanned system. Therefore, the performance and validation of the finished product characteristics required various, often unrelated tasks, whose success, however, determined the success of the entire undertaking. The research and engineering activities presented are certainly thematically diverse. However, the technological challenge undertaken here was in this case to develop and test a handy, inexpensive and relatively effective miniature unmanned system, which was to be built from ready-made elements commercially available. It was not about carrying out specialist tests and research focused on a single innovative device. The overarching goal was to build a system that, as serviceable and affordable, would have a chance for quick commercial implementation.

Key benefits gained from described work are the following:original UAV-construction demonstrating useful technological and aerodynamic qualities;effective adaptation of the MP2128g in terms of complex integration and successful setting all PID-loops;entry stage of mini-UAV propulsion system optimization, i.e., measuring values of thrust and effective power for selected battery configurations;elaborating self-made simple video-head useful for training and testing flights when high quality of the video imaging is rather secondary target.

The most useful and fully completed implementation seems to be the autopilot integration and successful tuning of its control loops. The development works on optimizing the mini-propulsion system and on constructing alternative versions of cheaper and easy-to-use video-head are being still carried out in the research team of the Institute of Aviation Technology of the MUT. The target mini-UAV version must be equipped with an avionics system fully integrated with the mission equipment, which will probably be an MP Dayview/Nightview camera. Preparation of the final prototype version requires additional operational tests; therefore, time and new financial outlays for the project are still necessary.

## Figures and Tables

**Figure 1 sensors-20-01752-f001:**
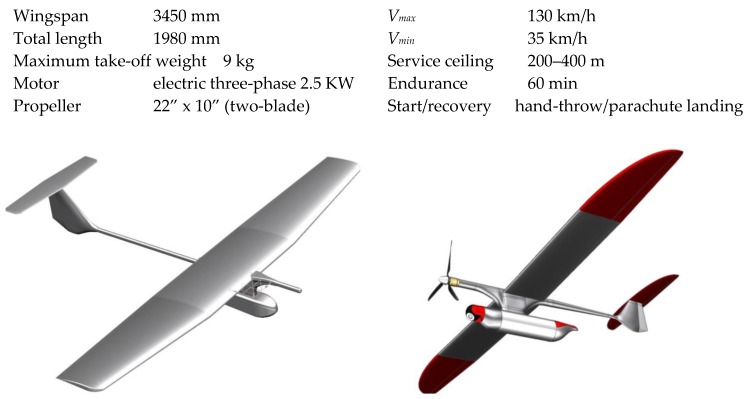
Developing stages of the mini-UAV CAD model—the initial conception and the finally developed configuration.

**Figure 2 sensors-20-01752-f002:**
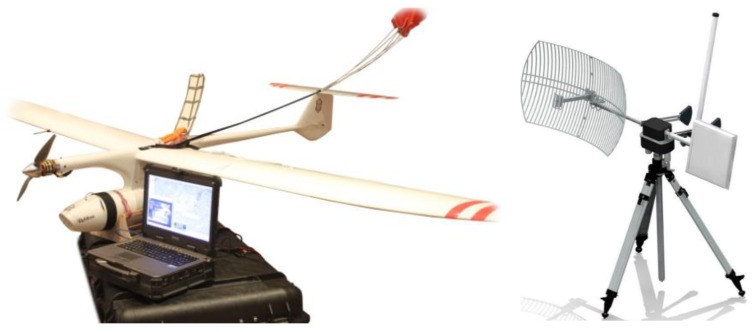
Unmanned aerial system (UAS) components: mini-UAV with the video head and recovery parachute, portable Ground Control Station—transported in a container, communication terminal.

**Figure 3 sensors-20-01752-f003:**
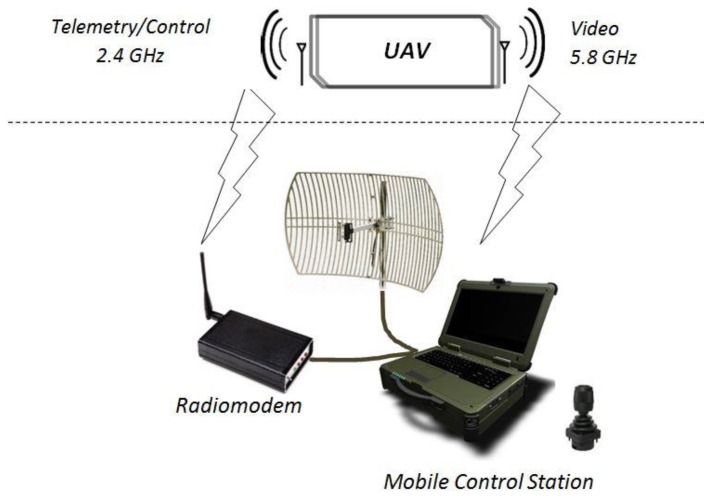
Conceptual view of the mini-UAS embracing the aerial component (mini-UAV) and parts of the on-ground component (GCS).

**Figure 4 sensors-20-01752-f004:**
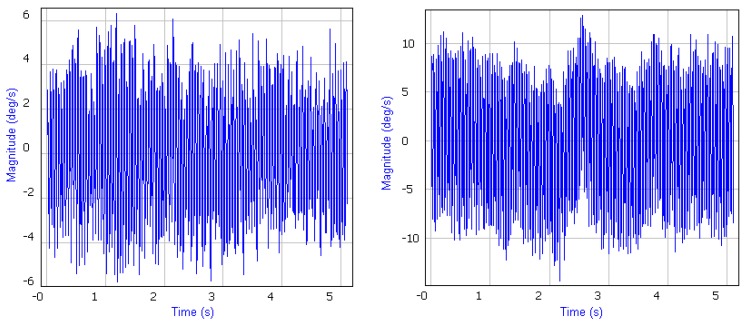
Gyroscope output during vibration tests.

**Figure 5 sensors-20-01752-f005:**
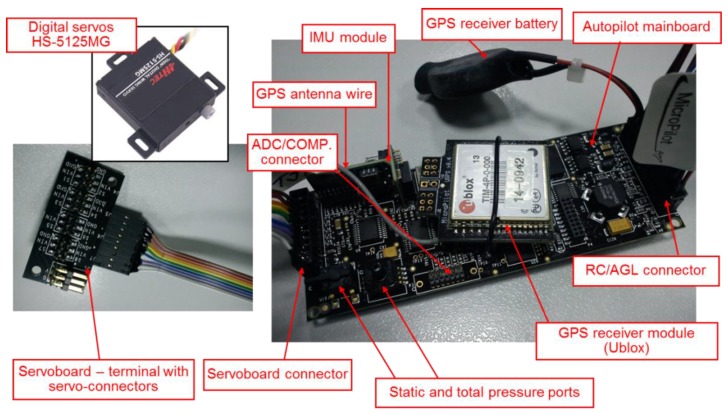
MP2128g autopilot main core and its primary peripheral elements.

**Figure 6 sensors-20-01752-f006:**
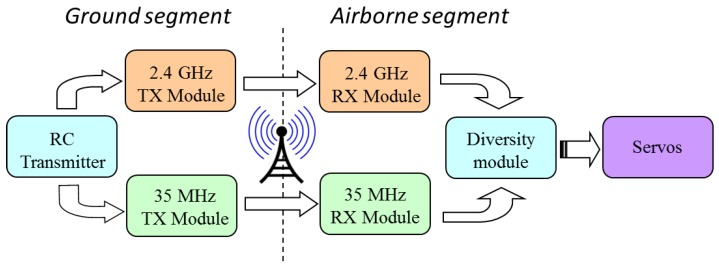
Block diagram of standard RC control system used in mini-airplane to test its flight qualities and to trim control surfaces—transmission redundancy was guaranteed by application two various transmission channels in frequencies: 33 MHz and 2,4 GHz; the diversity device was responsible for quick sampling, comparing and transceiving one of two signals with better quality.

**Figure 7 sensors-20-01752-f007:**
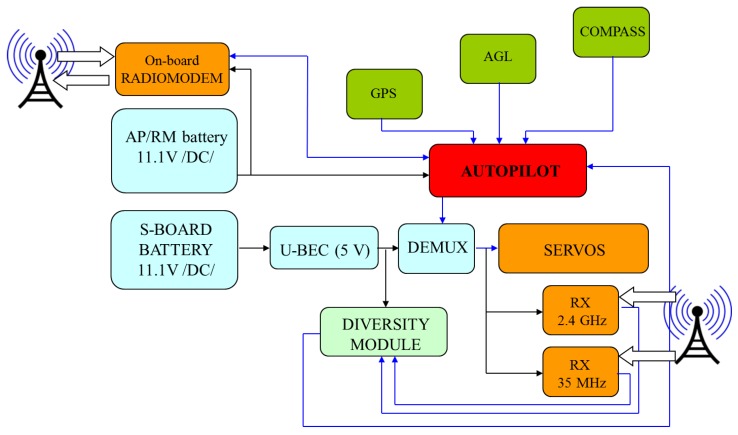
Block diagram of control system with MP2128g included–configuration allowing switch between manual mode (PIC–Pilot-in-Command) and automated mode (CIC–Computer-in-Command); in PIC mode remoted control is always supported by two independent transmissions.

**Figure 8 sensors-20-01752-f008:**
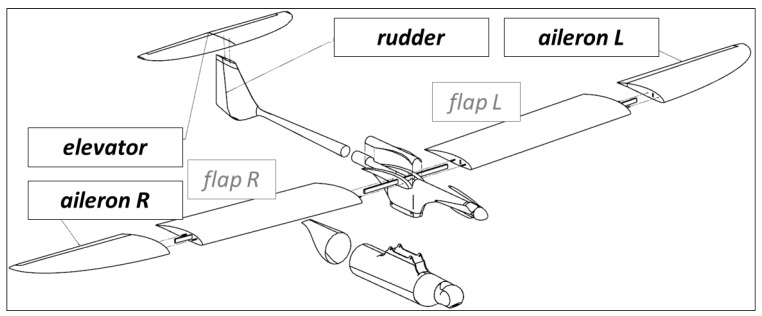
Control surfaces of the mini-UAV Rybitwa.

**Figure 9 sensors-20-01752-f009:**
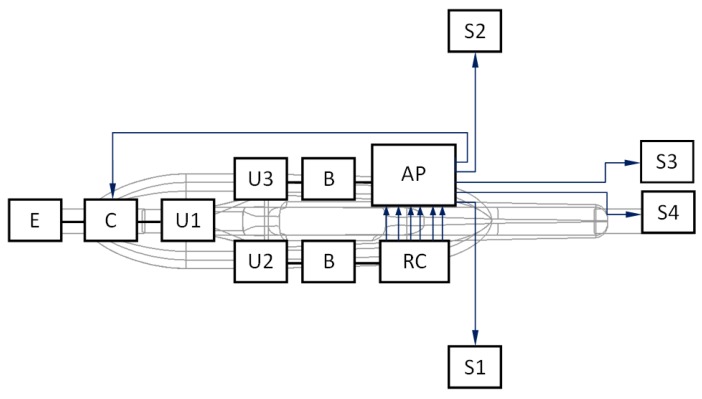
Block diagram of the avionics arrangement on the background of the fuselage chamber and the suspended pod: E, electric engine; C, electronic speed controller; U1, U2, U3, power sources (Li-Po type)—B, battery eliminator circuit (BEC)—AP, autopilot MP2128g—RC, receiver of remote radio-control signal—S1, S2, S3, S4, servomechanisms.

**Figure 10 sensors-20-01752-f010:**
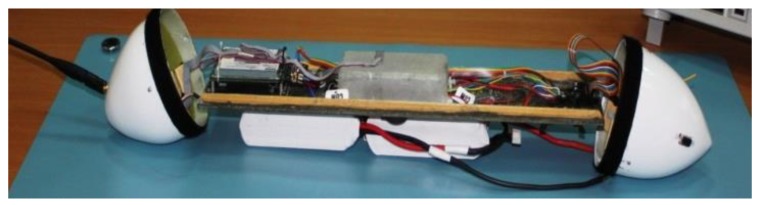
Mounting tray to be installed inside the pod with electronic devices and suspended battery packages; there are hemispherical chambers with additional elements on both sides of the container.

**Figure 11 sensors-20-01752-f011:**
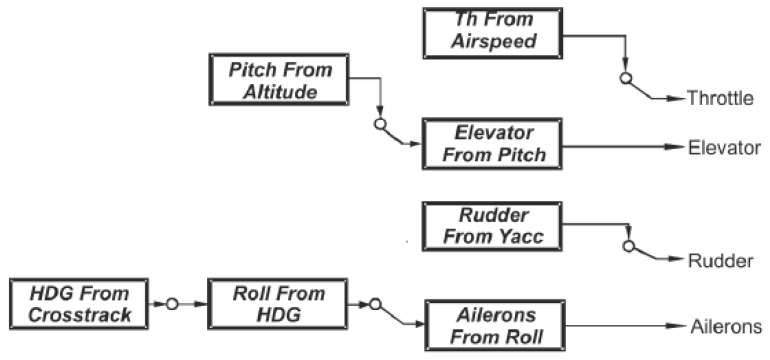
Scheme of enabled feedback loops during an exemplary level flight.

**Figure 12 sensors-20-01752-f012:**
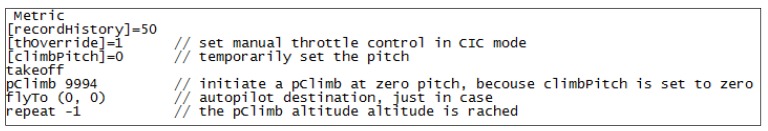
Section of the *.fly file with feedback loop set for climbing.

**Figure 13 sensors-20-01752-f013:**
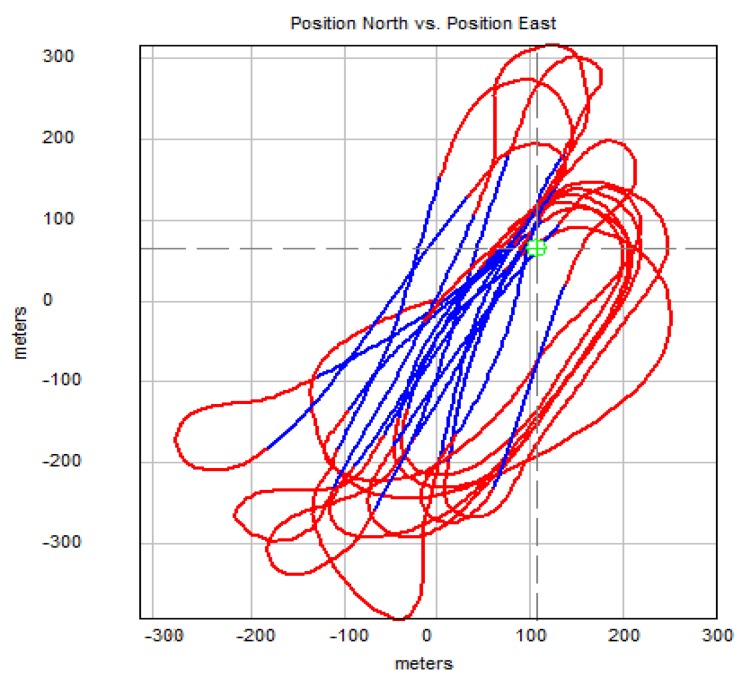
Flight path of mini-UAV received from logged coordinates—track sections performed in CIC mode are marked with blue.

**Figure 14 sensors-20-01752-f014:**
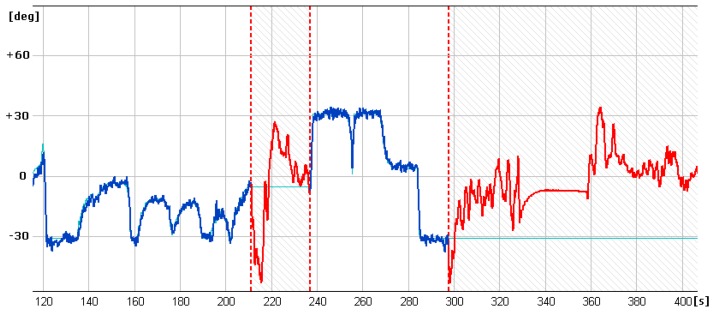
Curves of current roll angle and desired roll angle versus time.

**Figure 15 sensors-20-01752-f015:**
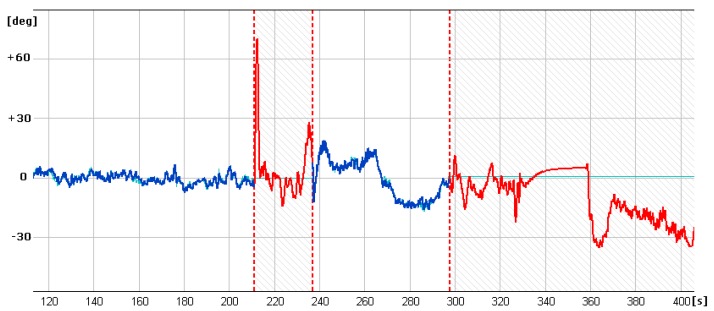
Curves of current pitch angle and desired pitch angle versus time.

**Figure 16 sensors-20-01752-f016:**
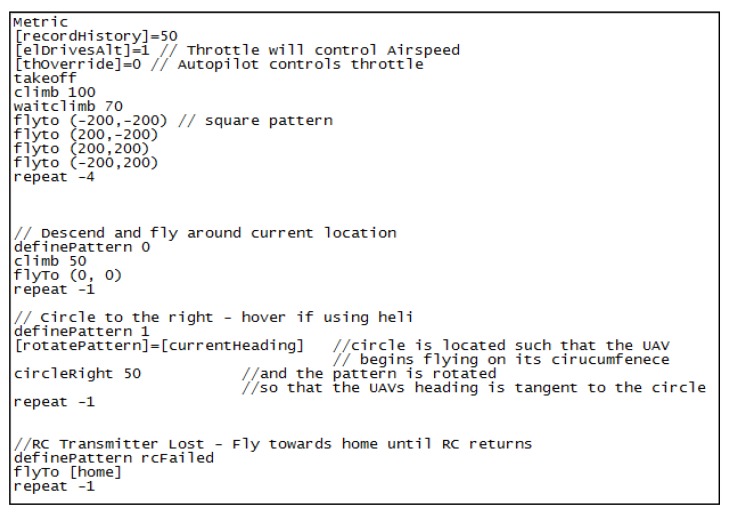
The content of exemplary *.fly file prepared for autonomous flight.

**Figure 17 sensors-20-01752-f017:**
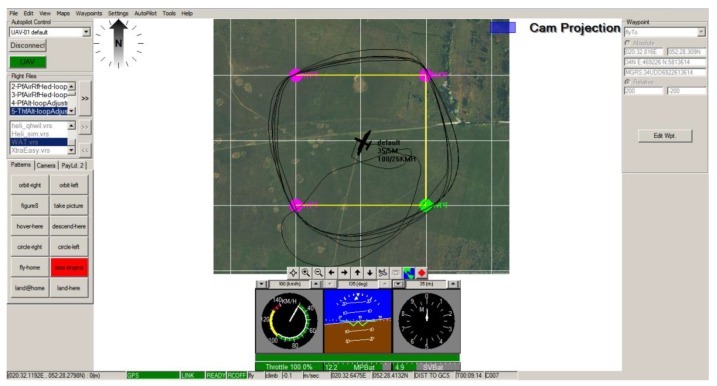
Exemplary view of the Horizon Software interface of the GCS with the path track referring to waypoints defined in previously set *.vrs file.

**Figure 18 sensors-20-01752-f018:**
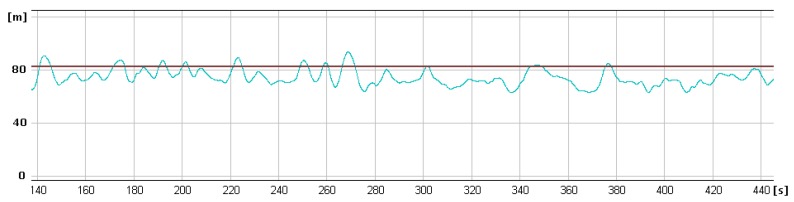
Currently estimated altitude (blue curve) and desired altitude (black horizontal line) in the autopilot control interval.

**Figure 19 sensors-20-01752-f019:**
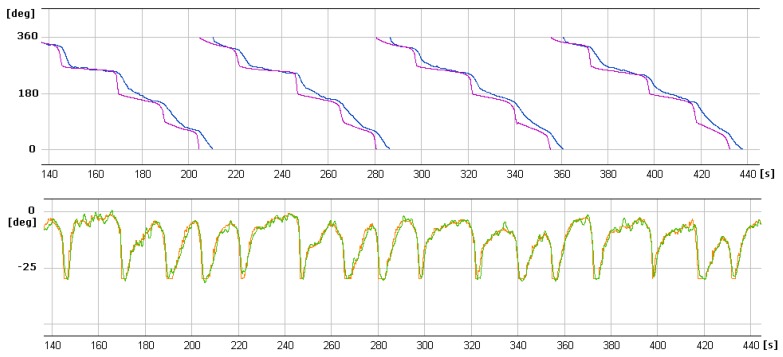
Currently estimated and desired runs of heading (top chart) and roll (bottom chart) angles.

**Figure 20 sensors-20-01752-f020:**
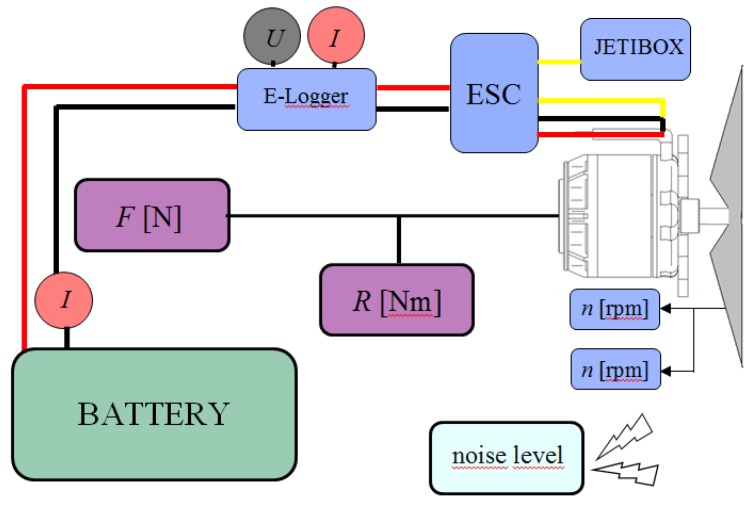
Scheme of measuring system installed on testbed for testing electric propulsion.

**Figure 21 sensors-20-01752-f021:**
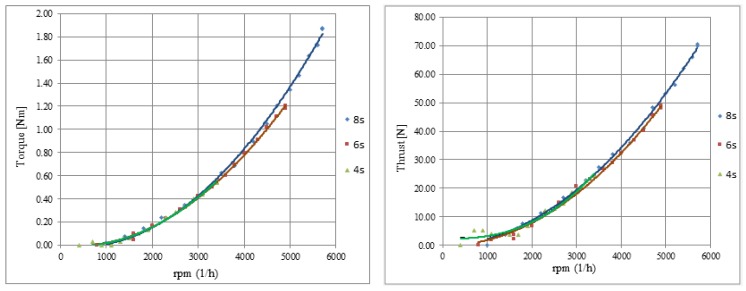
Graphs of torque and thrust versus rpm for three power sources (4 s/6 s/8 s).

**Figure 22 sensors-20-01752-f022:**
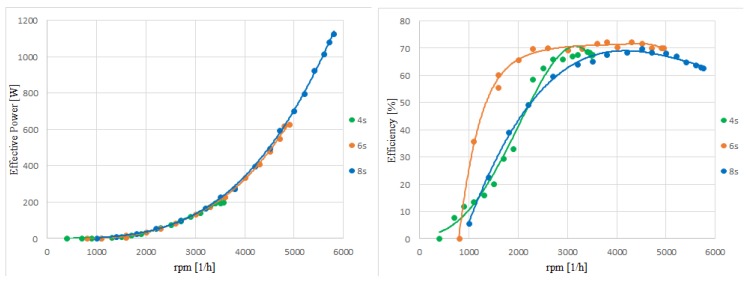
Graphs of effective power and efficiency (estimated as a ratio: P_eff_/P_elect_) versus rpm for three power sources (4 s/6 s/8 s).

**Figure 23 sensors-20-01752-f023:**
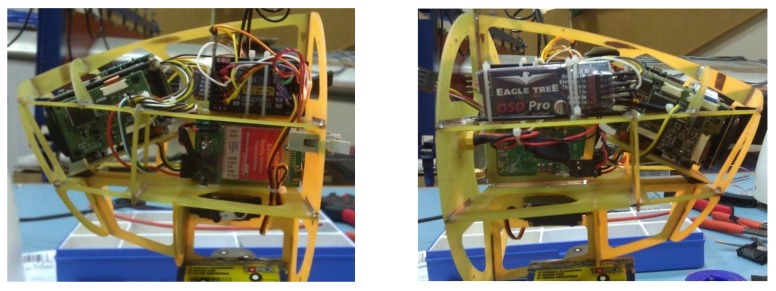
Lateral views of the video-head thin-walled construction with electronic items installed.

**Figure 24 sensors-20-01752-f024:**
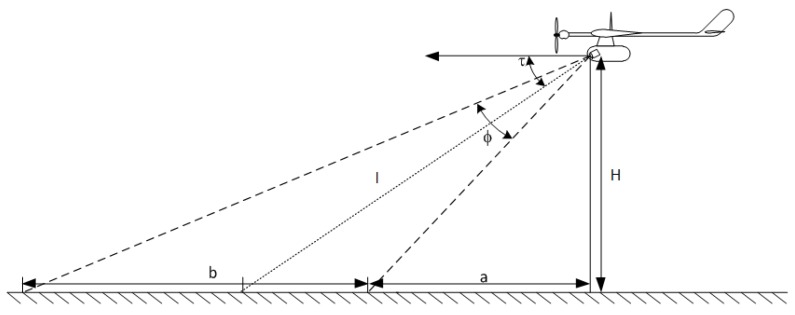
Scheme of geometric parameters for camera viewing in plane of flight trajectory.

**Figure 25 sensors-20-01752-f025:**
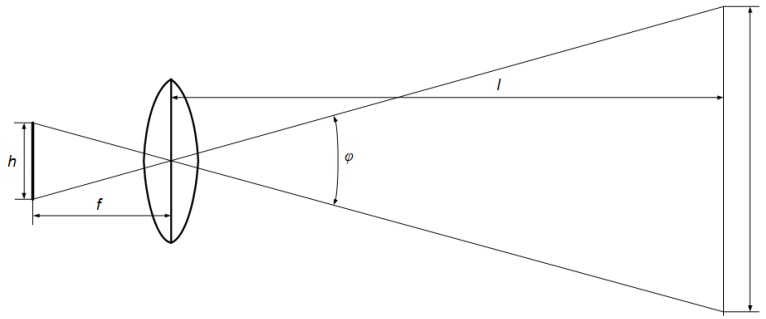
Simplified scheme of characteristic geometric parameters in side-view for relation between a CCD matrix, lens and an observed object.

**Table 1 sensors-20-01752-t001:** Extract of UAV categories defined by UVS-International, according to [1] or [2].

No	Category Name	Mass [kg]	Range [km]	Flight Altitude [m]	Endurance [h]
1	Micro	<5	<10	250	1
2	Mini	<25/30/150	<10	150/250/300	<2
3	Close Range	25–150	10–30	3000	2–4
4	Medium Range	50–250	30–70	3000	3–6
5	High Alt. Long Endurance	>250	>70	>3000	>6

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
