# Peer review of "Integration and Investigation of Selected On-Board Devices for Development of the Newly Designed Miniature UAV"

_sensors, 2020, doi:10.3390/s20061752_

Round 1

Reviewer 1 Report

  This submission deals with crucial technical topics realting design, complex integration and development process of mini-UAV.  Discussions and analysis have well covered essential issues including function definition, hardware configuration, system architecture, component integration and capability verifications. These research and test results have demonstrated useful technological explorations and commercial references.  

  Suggestion would be made to authors that the whole prototype version should be further tested under diffrent dynamical conditions like loading, speed in contrast to attitude and spatial performances.

Reviewer 2 Report

In this paper authors present results from what it seems to be a very interesting research project, which consists on the design of a small autonomous airplane devoted for inspection. Even though there is not much novelty in the project, I acknowledge that there are several related technological challenges. It happens, however, that there is no clear discussion in the paper related to such challenges.

In fact, the paper has too many problems. It appears to be more like a technical report than a proper research paper. For instance:

There is no proper presentation of the problem under consideration. It also lacks an overview of the proposed solutions.

There are no discussions about related works. There are plenty of other research projects devoted for UAV design with similar technological challenges. This must be better explored.

The technological challenges are not properly addressed in the paper. For instance, aren’t there any communication challenge between controller and devices? I do not believe it is just a plug-and-play task.

Presented results are uninteresting the overall research community – at least in the way they are presented and in the form that the paper is organized. Authors should properly define what is a "miniature aircraft" (bellow how many gramms: 2g, 200g, 2000g?). And who (which authority organization) provides such definition?

A clear, formal, overview of the related research project should be provided.

Reviewer 3 Report

Nice technical paper.

p.9. Fig. 13, too small font, x , y axis not labeled

p.9, Fig. 14, too small font, y axis not labeled

p.10. Fig. 17, too small font, x , y axis not labeled

p.11. Fig. 18, too small font, x , y axis not labeled

Keep the same style as e.g.  Fig. 21

p. 11, r.279 (Fig. 17-18) Fig -> Figs

Author Response

Dear Sir/Madam

Indeed the Figures 13, 14, 17, 18 have to be corrected as fonts are too small and axes are not labelled. Of course required corrections will be introduced into the paper. The style kept in Figs. 17-18 is much more better as those graphs were prepared in Excel while former graphs were generated directly from some embedded-in-system application viewing flight data logs. Probably some more graphs will be added to the paper as the other Reviewer suggests to explain some aspects more precisely to discuss properly technological challenges taken in the work.

Round 2

Reviewer 2 Report

Even though authors performed positive modifications in the paper to improve its quality, it is my understanding that it does not match the requirements to be accepted as a journal paper. Follows my strongest concerns towards acceptance:

  • There is no proper presentation of the problem under consideration. It also lacks an overview of the proposed solutions.

In authors’ response letter, it is claimed that the paper focus on the “presentation of the autopilot integration and tuning process”. But what are the research challenges related with such integration? Did any other research team already addressed such challenge? What are the possible solutions? For instance, section 4.1 convers part of authors claim in a very poor manner. It does not contain any contribution for the research community.

  • There are no discussions about related works.

In this revised version authors simply extended the list of bibliographic references. It needs, however, more discussion: the problems under consideration in the paper should be confronted with solutions provided by such related works.

Author Response

We have improved the manuscript trying to adapt to the stated questions and expectations. The changes were introduced in principle in point 3 and slightly in point 4. The list of References was a little extended also, as some other positions were cited in publication overview.

We tried to highlight the context of our work by referring to the indicated publications regarding the application and testing of autopilots. Challenges related to the tasks undertaken in our work may be of a project-technological or purely scientific nature. Our work was primarily focused on designing and manufacturing a miniature UAV according to our original idea and then equipping it with avionic items available on the market. The main goal was to obtain a system demonstrator that meets the basic design assumptions imposed on the mini-UAV class system. In our undertaking, the technological aspect was the most important, which means the development of a complete efficient system. Adapting and tuning the MP2128 autopilot was the most difficult part task in the process of system development, but not the only one. Other tasks - such as airframe construction, equipment installation, propulsion unit selection, constructing training video-head were also very important in the context of the project's success. However, it was not our goal to develop new avionics elements such as transducers, sensors and transmitters. It was also not about testing and investigating individual elements, analyzing the sensitivity of these elements to the impact of possible design or operational factors. For this reason, the scientific effects of our work may not be visible too much, but the engineering and application effects should certainly be noticeable.